# Anti-Hair Loss Effect of Adenosine Is Exerted by cAMP Mediated Wnt/β-Catenin Pathway Stimulation via Modulation of Gsk3β Activity in Cultured Human Dermal Papilla Cells

**DOI:** 10.3390/molecules27072184

**Published:** 2022-03-28

**Authors:** Jaeyoon Kim, Jae Young Shin, Yun-Ho Choi, Nae Gyu Kang, Sanghwa Lee

**Affiliations:** LG Household & Health Care (LG H&H) R&D Center, 70, Magokjoongang 10-ro, Gangseo-gu, Seoul 07795, Korea; kjy5281@lghnh.com (J.K.); sjy2811@lghnh.com (J.Y.S.); youknow@lghnh.com (Y.-H.C.)

**Keywords:** adenosine, Wnt/β-catenin signaling, adenosine receptor, gsk3β, anti-hair loss

## Abstract

In the present study, we investigated the molecular mechanisms of adenosine for its hair growth promoting effect. Adenosine stimulated the Wnt/β-catenin pathway by modulating the activity of Gsk3β in cultured human dermal papilla cells. It also activated adenosine receptor signaling, increasing intracellular cAMP level, and subsequently stimulating the cAMP mediated cellular energy metabolism. The phosphorylation of CREB, mTOR, and GSK3β was increased. Furthermore, the expression of β-catenin target genes such as Axin2, Lef1, and growth factors (bFGF, FGF7, IGF-1) was also enhanced. The inhibitor study data conducted in Wnt reporter cells and in cultured human dermal papilla cells demonstrated that adenosine stimulates Wnt/β-catenin signaling through the activation of the adenosine receptor and Gsk3β plays a critical role in transmitting the signals from the adenosine receptor to β-catenin, possibly via the Gαs/cAMP/PKA/mTOR signaling cascade.

## 1. Introduction

Adenosine is one of the purinergic nucleosides that participate in diverse cellular and physiological processes. Therefore, adenosine and its derivatives are used as therapeutics for cardiac function, immune system, and central nervous system [1,2]. For example, adenosine is used as a prescription medicine for paroxysmal supraventricular tachycardia (PSVT) by intravenous injection [3,4]. Topical administration of adenosine is under clinical investigation for wound-healing deficiency [5]. In addition, an anti-hair loss effect of adenosine was also reported, improving androgenetic alopecia (AA) and female-pattern hair loss (FPHL) by topical application [6,7,8]. Adenosine was reported to stimulate the expression of fibroblast growth factor 2 and 7 (FGF2 and FGF7) in cultured dermal papilla cells (DPCs), which are responsible for hair growth via adenosine A_2B_ activation [9,10]. The precise molecular mechanisms of adenosine for promoting hair growth, however, are not clearly elucidated yet.

One of the major therapeutic targets for alopecia treatment is the Wnt/β-catenin signaling pathway. The regulation of the Wnt/β-catenin pathway is important in the hair follicle cycle and its control [11]. Activation and downregulation of the β-catenin pathway are critical for maintaining the anagen stage and for anagen to catagen transition, respectively [12]. Furthermore, Wnt signaling plays critical roles in telogen–anagen transition of the human hair cycle, activating the quiescent hair follicle stem cells [13].

Activation of the adenosine receptor could contribute to diverse phenotypes and pathways. One of the stimulated pathways is the Wnt/β-catenin pathway. In osteoblast and neuronal cells, adenosine receptor activation triggers the Wnt/β-catenin pathway [14,15]. Furthermore, in our previous study, we found that adenosine stimulated the Wnt/β-catenin pathway through the modulation of Gsk3β activity in human dermal fibroblasts [16]. The Gsk3β has been suggested as a connecting point between adenosine receptor signaling and β-catenin activation in human dermal fibroblasts.

In the present study, we investigated the molecular mechanisms of adenosine for its hair growth stimulating activity, elucidating the relationship between adenosine receptor activation, and Wnt/β-catenin signaling in cultured human dermal papilla cells (hDPCs).

To do this, the expression pattern of adenosine receptor subtypes, A_1_, A_2A_, A_2B_, and A_3_ in hDPCs was examined and the effects of adenosine treatment on Wnt/β-catenin signal activation were scrutinized. To clarify which signal transduction pathways are involved, we investigated the changes in the phosphorylation of cell signal transduction elements upon adenosine stimulation such as various MAP kinases, Gsk3β, mTOR, and CREB in cultured hDPCs. In addition, the effects of adenosine on intracellular cAMP level, cAMP mediated energy metabolism, and the expression of Wnt target genes such as growth factors (FGF2, FGF7, and IGF-1) and transcription factor (LEF1) were also investigated. To understand the more precise mechanism of Wnt/β-catenin pathway stimulation by adenosine, the effects of various inhibitors (ZM241385, PBS603, KH7, H89, LY294002, and IWR1) on adenosine induced Wnt activities were investigated.

## 2. Results

### 2.1. Adenosine Activated Wnt/β-Catenin Pathway in Cultured hDPCs

In our previous report, adenosine activated the Wnt/β-catenin signaling pathway through the modulation of Gsk3β activity [16]. To elucidate the molecular mechanism of the hair growth promoting activity of adenosine, the effects of adenosine on the Wnt/β-catenin signaling pathway were investigated in cultured hDPCs. Cells were treated with adenosine and recombinant Wnt3a, a positive control, and the protein levels of β-catenin, Gsk3β, and phospho-Gsk3β were analyzed at different time points (0 to 240 min) to compare the time course of signal activation. As shown in Figure 1a, adenosine was found to activate the Wnt/β-catenin signaling pathway in cultured hDPCs. The β-catenin expression was significantly increased at 30 min after adenosine treatment. The inhibitory phosphorylation level of Gsk3β-Ser9 site was elevated as soon as 1 min after treatment and lasted for 15 min (Figure 1b), and gradually diminished to basal level at 30 min treatment.

The extracellular adenosine in a tissue is detected by adenosine receptors. The distribution of four adenosine receptor subtypes, A_1_, A_2A_, A_2B_, and A_3_, varies with cell types [1]. It was revealed that the expression patterns of adenosine receptor subtypes in HEK293A reporter cells and in hDPCs were similar; A_2B_ was dominant and A_2A_ ranked second. The A_3_ type was not detected, and the A_1_ type was expressed less than 5% of A_2A_ (Figure 1c). Therefore, it could be presumed that the consequences of adenosine treatment in HEK293A cells and hDPCs are exerted predominantly through the activation of the adenosine A_2A_ and A_2B_ receptors.

### 2.2. Adenosine Enhanced Cell Viability and Mitochondrial Energy Metabolism in Cultured hDPCs

Since adenosine plays critical roles in cellular energy production and cell survival, the effects of adenosine on these functions were investigated. As shown in Figure 2a, cellular cAMP level was significantly elevated by adenosine treatment in cultured hDPCs. We also found that the cell viability, measured by NAD(P)H generation, was significantly increased by adenosine treatment for 4 h (Figure 2b). Maintaining the energy metabolism at a higher level is essential for the elongation of the anagen stage of hair follicles [17]. In this aspect, our data demonstrate that the enhanced cell proliferation with increased energy metabolism by adenosine in hDPCs could partly explain the mechanism of adenosine for its hair growth promoting effect.

Mitochondrial membrane potential (△Ψ), a major marker of activated mitochondrial function, was also increased, being the possible driving force for the enhanced energy metabolism by adenosine (Figure 2d). In addition, we investigated the mRNA levels of cell proliferation and cell cycle related markers. The expression level of Ki67, a cell proliferation marker, was significantly elevated by adenosine in a concentration dependent manner. The mRNA expression levels of p21 (CDKN1A) and p16 (CDKN2A), the cell cycle arrest markers, in contrast, were significantly decreased in concentration dependent manners (Figure 3).

### 2.3. Adenosine Increased the Phosphorylation of CREB, mTOR, and Gsk3b in Cultured hDPCs

To clarify how adenosine stimulate Wnt/β-catenin signaling pathway, the phosphorylation of CREB and various protein kinases including MAPKs after adenosine treatment for 10 min was investigated in cultured hDPCs. The phosphorylation of CREB, p38, Gsk3β, mTOR, and p70S6K was significantly increased by adenosine treatment in cultured hDPCs (Figure 4). The increased phosphorylation of Gsk3β, mTOR, and p70S6K is consistent with those observed in human dermal fibroblasts in our previous report [16]. The increased phosphorylation of CREB, on the other hand, might be responsible for the enhanced energy metabolism by adenosine since cAMP and PKA, a downstream target of cAMP [18], are strongly correlated with energy metabolism, especially in mitochondria [19].

### 2.4. Adenosine Activated Wnt Target Genes and Growth Factors in hDPCs

To elucidate the effect of adenosine on the Wnt/β-catenin signaling pathway in hDPCs, we evaluated the protein and mRNA expression levels of the Wnt/β-catenin signaling targets. The β-catenin expression level was increased by adenosine treatment for 1d (Figure 5a). As a consequence, versican, a major anagen marker of DPCs [20], whose expression level is controlled by β-catenin/TCF [21], was also increased in a concentration dependent manner (Figure 5a).

The mRNA expression of AXIN2, Lef1, and several growth factors was also examined after treatment with adenosine for 24 h. The AXIN2 expression level, one of the Wnt target genes [22], was enhanced by adenosine treatment. Adenosine also significantly increased the expression level of Lef1 (Figure 5b), one of the major β-catenin responsive genes, which encodes transcription factor and stimulates the anagen hair follicle stage [23]. The expression of FGF2, FGF7 and IGF-1, previously reported to be stimulated by Wnt [24,25] and to promote hair growth and elongate anagen stage [26,27], was enhanced by adenosine treatment (Figure 5c). Our data demonstrate that the activation of the Wnt/β-catenin pathway through the adenosine receptor activation could be a plausible molecular mechanism for the hair growth promoting activity of adenosine.

### 2.5. Adenosine Mediated-Wnt/β-Catenin Signaling Was Inhibited by Adenosine Receptor Antagonists

It was revealed that adenosine and NECA, adenosine receptor agonists, increased the luciferase activity of the WRHEK293A reporter cell line, which was abrogated by the treatment of IWR1, an Axin complex inhibitor, comparable with that of Wnt3a (Figure 6). Our data suggest that adenosine and NECA stimulate the canonical Wnt/β-catenin signaling pathway and the APC/Axin/Gsk3β destruction complex is engaged (Figure 6). The activated Wnt signaling was abolished by the treatment of ZM241385 and PSB 603, adenosine A_2A_ and A_2B_ inhibitors, respectively [28], implicating the involvement of adenosine receptor A_2A_ and A_2B_.

Adenosine A_2A_ and A_2B_ receptors are a family of G-protein coupled receptors. The A_2A_ subtype is highly associated with the Gα_s_ subunit and the A_2B_ subtype has higher affinity to both the Gα_s_ and Gα_q_ subunits [18]. It has previously been reported that the stimulation of Gα_s_ subunits of adenosine receptors activates adenylyl cyclase [29]. As shown in Figure 7, the induction of Wnt/β-catenin signaling by adenosine and NECA was decreased by the adenylyl cyclase inhibitor KH7. Our data suggest that adenylyl cyclase plays a pivotal role in transmitting the signals from the adenosine receptor to Wnt/β-catenin signaling.

cAMP is one of the major second messengers for triggering various physiological changes such as proliferation, differentiation, and migration [30]. PKA is a well-known downstream target of cAMP [31]. Because our data showed that adenylyl cyclase played an essential role in adenosine-mediated Wnt/β-catenin signal activation and cAMP was activated by adenosine treatment in DPCs (Figure 2a), the effect of H89, a PKA inhibitor, was examined. As shown in Figure 7, the adenosine induced Wnt signaling was aborted by H89. In addition, LY294002, a broad spectrum inhibitor of PI3K, mTOR, and CK2 [32] also inhibited adenosine- and NECA-induced Wnt signaling. Taken together, adenosine-mediated Wnt/β-catenin signaling pathway activation depends on the activities of PKA and mTOR (Figure 7).

### 2.6. Adenosine Induced β-Catenin Activation Depends on Adenosine Receptors and PKA Activity in Cultured hDPCs

To confirm whether the similar molecular pathway also works in cultured hDPCs, the effects of adenosine receptor and PKA inhibitors on adenosine induced β-catenin activation were examined. The increased β-catenin level by adenosine treatment was decreased in the presence of PSH603 or ZM241385. In addition, the treatment of H89, a PKA inhibitor, also downregulated β-catenin protein level, elevated by adenosine (Figure 8).

Our data suggest that adenosine A_2A_ and A_2B_ receptors and PKA are also indispensable in the stimulation of Wnt/β-catenin signaling by adenosine in cultured hDPCs.

By summarizing the inhibitor study data, the adenosine mediated Wnt/β-catenin signaling activation was correlated with adenylyl cyclase and PKA activity, depending on mTOR activities. Therefore, our data demonstrate that adenosine stimulated Wnt/β-catenin signaling through modulating the Gsk3β activity by inhibitory phosphorylation and suggest that the signaling cascade of cAMP, PKA, and mTOR likely plays essential roles in this process.

## 3. Discussion

In the present study, we investigated the cellular and molecular effects of adenosine in cultured hDPCs, elucidating the signal transduction pathways for the hair growth promoting activity of adenosine.

Adenosine receptors belong to the superfamily of G protein-coupled receptors (GPCRs), which bind to the heterotrimer of guanine nucleotide-binding proteins, comprising GDP-bound Gα, Gβ, and Gγ subunits [33]. The 16 Gα subunits were identified and can be classified into four types: G_s_, G_i/o_, G_q/11_, and G_12/13_ [29]. The G_s_ binds and activates adenylyl cyclase (AC), triggering protein kinase A (PKA) activity via cAMP. On the other hand, the G_q/11_ subunit binds and activates phospholipase Cβ (PLCβ), activating protein kinase C (PKC) and phosphoinositide 3-kinases (PI3K). Our data suggest that the adenosine stimulated Wnt/β-catenin signaling pathway in hDPCs depends on adenylyl cyclase activity stimulated by the Gα_s_ subunit of the adenosine receptor.

Topical application of adenosine has been used to promote hair growth and skin health [34,35]. The hair follicle is one of the most energy consuming organs in humans. Therefore, the energy expenditure is higher, especially in the anagen stage [17]. In this context, maintaining the energy metabolism in hair follicles at higher rates could be beneficial for the prevention and/or improvement of alopecia. For example, enhanced mitochondrial potential in ex vivo experiments with human hair follicles was reported to stimulate hair follicle growth and prevent anagen–catagen transition [36,37,38]. Since mitochondrial energy generation in hDPCs accounts for the majority of the cells’ energy metabolism [17], the induction of mitochondrial activity might drive the growth of human hair both in vitro and in vivo [39,40].

We found that adenosine treatment in cultured hDPCs increased NADH generation and mitochondrial membrane potential (△Ψ), which are strongly connected with the positive steady state of NAD/NADH coupling (Figure 2b,d). Furthermore, the cAMP level and the phosphorylation of CREB were enhanced by adenosine (Figure 2a and Figure 4). It was reported that cAMP mediated PKA and CREB activation promoted mitochondrial gene expression and cell survival via mitochondrial enhancement [19,41]. As consequences, the mRNA level of Ki67, a proliferation marker, was increased, and the senescence markers, CDKN1A and CKDN2A, were decreased by adenosine treatment in cultured hDPCs (Figure 3). Therefore, our data suggest that adenosine consecutively stimulates adenosine receptor, cAMP level, CREB, and mitochondrial energy metabolism. This adenosine induced mitochondrial activation also might be beneficial to the prolongation of the hair follicle anagen stage.

The growth factors, secreted by DPCs, control the hair follicle cycle [42]. FGF7 is one of the important cytokines for the induction of a new hair follicle cycle and hair germ proliferation [43] and was proposed as the main mechanism for adenosine’s hair growth promoting activity [15]. Consistent with this previously reported data, adenosine significantly increased the mRNA expression levels of bFGF, FGF7, and IGF-1 (Figure 5c). It could be predicted that the adenosine-mediated growth factor supply to the hair bulb at a higher level could contribute to prolong the anagen period and support hair growth.

The adenosine A_2A_ and A_2B_ receptor signaling pathways have been suggested as potential targets for alopecia treatment [44] and proposed as one of the mechanisms of action for minoxidil in improving alopecia, regulating the adenosine level via SUR2B receptor activation in DPCs [45]. We have previously reported that adenosine and its natural derivative activated the Wnt/β-catenin pathway and stimulated cell migration in human dermal fibroblasts [16] through the activation of protein kinases such as MEK1/2, mTOR, and p70S6K and the inhibitory phosphorylation of the Gsk3β Ser9 site wase suggested as key mechanisms for adenosine mediated Wnt/β-catenin pathway activation, interconnecting the adenosine receptor activation and Wnt/β-catenin signaling.

In the present study, we also found that adenosine activated Wnt/β-catenin signaling by modulating the activity of Gsk3β in cultured hDPCs was consistent with our previously reported data. Although β-catenin could be stabilized independent of the phosphorylation status of GSK3β [46,47], inhibitory serine-phosphorylation is the most frequently examined mechanism that regulates the activity of GSK3 in Wnt signal activation [48,49]. Our data demonstrate that the inhibitory ser-9 phosphorylation is a major determinant in adenosine induced Wnt signal activation, but we cannot rule out the possibility that the GSK3β independent pathway also worked together. We focused on investigating the molecular signaling elements between the adenosine receptor and Gsk3β. Several protein kinases (p38, Gsk3β, mTOR, and p70S6K) and CREB were significantly activated by adenosine treatment and these elements would play important roles in the adenosine-mediated Wnt/β-catenin signaling cascade (Figure 4). Since the phosphorylation of Gsk3β-S9 was reported to depend on the PKA and PI3K/AKT/mTOR pathways [50], we examined the effects of various inhibitors for adenosine receptors (ZM241385 and PBS603), MAP kinases (KH7, H89, LY294002), and IWR-1 on Wnt reporter activities. We found that KH7 (adenylyl cyclase inhibitor), H89 (PKA inhibitor), and LY294002 (mTOR inhibitor) significantly downregulated the adenosine induced Wnt/β-catenin activities (Figure 7). In addition, we further investigated the adenosine induced β-catenin levels in the presence of two adenosine receptor antagonists and PKA inhibitor (ZM241385, PBS603 and H89) in cultured hDPCs, demonstrating that adenosine activated Wnt/β-catenin signaling in cultured hDPCs also depends on adenosine receptors and PKA activity (Figure 8).

In this research, we used various kinase inhibitors for elucidating the roles of several protein kinases in adenosine induced Wnt signaling activation. Some inhibitors were reported to inhibit not only the kinases of concern, but also some other ones. For example, H89 inhibited not only PKA, but also RSK1 and MSK1 [51]. LY294002 inhibited PI3K, mTOR, and CK2 [32]. Despite the off target effects caused by the broad specificities of some inhibitors, we could deduce the roles of specific kinases by focusing on the kinases previously reported to participate in adenosine induced signal transduction cascades. Furthermore, knock-out and/or knock-down of specific protein kinases could help to elucidate precise signaling cascade by adenosine treatment

We treated adenosine in cultured hDPCs at concentrations of µM to mM ranges, much higher than physiological adenosine concentrations. Extracellular adenosine concentration in normal cells is approximately 300 nM; however, in response to cellular damage, the adenosine concentration is rapidly elevated to the µM level to trigger tissue protection and repair [52]. The affinities of adenosine to specific adenosine receptor subtypes were previously reported in CHO cells, showing the EC_50_ values for adenosine receptor subtypes A_1_, A_2A_, A_2B_, and A_3_ as 310 nM, 700 nM, 24,000 nM, and 290 nM, respectively [5]. Because the concentrations of adenosine used in this study exceeded the EC_50_ of the least sensitive receptor A_2B_, the results might be the consequences of harmonized responses of multiple adenosine receptor subtypes, A_1_, A_2A_, and A_2B_ (Figure 1c). In addition, clarifying the contribution rate of each of the subunits G_s_, G_i/o_, and G_q/11_ and further study with adenosine receptor knock-out and Gα subunit inhibitors (cholera toxin, pertussis toxin, and UBO-QIC) will be helpful to understand the more precise mechanism of adenosine and to elucidate the effective therapeutic targets of alopecia treatment [53,54].

When adenosine is used as a therapeutic, there could be concerns about adverse effects, especially systemic ones, since adenosine has profound effects on the heart. We focused on the scalp, which is mostly the external body site and is the route for topical application. Although the topically applied chemicals could enter the circulation, topical application would be a good regimen to circumvent the systemic adverse effects. Furthermore, the plasma half-life of adenosine was reported to be extremely short, less than 10 s with intravenous administration. Based on these considerations, the topically applied adenosine should not provoke significant systemic adverse effects but this aspect deserves further investigation.

In conclusion, our data demonstrate that adenosine simulates the Wnt/β-catenin signaling pathway through adenosine receptor activation by modulating the activity of Gsk3β, and the Gα_s_/cAMP/PKA/mTOR cascade plays a pivotal role in adenosine mediated Wnt pathway activation, being the underlying molecular mechanism of adenosine for its hair growth promoting effect (Figure 9).

## 4. Materials and Methods

### 4.1. Dermal Papilla Cell Culture

hDPCs were purchased from Promocell (Heidelberg, Germany) and cultured in basal medium supplemented with 4% fetal calf serum, 0.4% bovine pituitary extract, 1 ng/mL basic fibroblast growth factor, and 5 μg/mL insulin (Supplement Mix, Promocell). Cells were maintained in a humidified incubator at 37 °C, 5% CO_2_. Adenosine stock solution (10 mM) was prepared just before each experiment. Before adenosine (Sigma-Aldrich, MO, USA) treatment, serum limitation was conducted by replacing the medium with fresh DMEM (Gibco, Waltham, MA, USA) supplemented with 1% FBS (Gibco, Waltham, MA, USA) and 1 ng/mL bFGF (Merck, Darmstadt, Germany) and culturing for 24 h to minimize the effects of serum and growth supplements. ZM241385, PBS603, KH7, H89, LY294002, and IWR1 were purchased from Tocris Bioscience (Bristol, UK).

### 4.2. Wnt Reporter Assay

WRHEK293A cells (Amsbio, Abingdon, UK) were seeded in a black 96-well plate and cultured for 24 h. Cells were treated with various concentrations of samples and incubated for another 24 h. Cells were then lysed by adding 50 μL of 1x passive lysis buffer (Promega, Madison, WI, USA) to each well and shaken for 10 min. GFP expression (internal cell viability control) was assessed by measuring the fluorescence at a 488/510 nm wavelength using VICTOR3 (PerkinElmer, Waltham, MA, USA). A total of 50 μL of luciferase substrate solution (Promega, Madison, WI, USA) was added and luciferase activity was measured using VICTOR3. Luminescence (TCF/LEF activity) values were normalized to GFP (cell viability) values.

### 4.3. Cell Viability Assay and Mitochondrial Membrane Potential Assay

The effect of adenosine on the viability of hDPCs was examined using the CCK-8 assay (Dojindo, MA, USA) and JC-1 mitochondrial membrane potential assay (Abcam, Cambridge, UK) kits following the manufacturer’s protocols. To examine the cellular energy metabolism, NAD(P)H generation was measured by the CCK-8 assay. The absorbance at 450 nm was read using a micro-plate reader (BioTek, Winooski, VT, USA). The mitochondrial membrane potential was measured by JC-1 staining. Briefly, after adenosine treated DPCs were stained with 1 μM JC-1 solution, fluorescence intensities from the JC-1 aggregate and monomer form were measured at 590 nm (535 nm excitation) and 530 nm (475 nm excitation), respectively, with Wallac Victor3 1420 (PerkinElmer, Waltham, MA, USA). Mitochondrial membrane potential (△Ψ) was visualized by taking fluorescence images with an EVOSTM FL Auto2 Imaging System (ThermoFisher scientific, Waltham, MA, USA).

### 4.4. Quantitative Real-Time PCR

Adenosine was treated at concentrations of 0.75, 1.5, and 3 mM for 24 h, with non-treated cells serving as the control. Total RNA was extracted using an Rneasy RNA Extraction Kit (Qiagen Inc., Hilden, Germany). cDNA synthesis was performed using a cDNA Synthesis Kit (Phillkorea, Seoul, Korea) with ThermoCycler (R&D systems, Minneapolis, MN, USA), according to the manufacturer’s protocols. cDNA samples obtained from the control and treated cells were subjected to real-time (RT)-PCR analysis.

TaqMan probes for RT-PCR used in this study were as follows: GAPDH assay id 4352934E; KI67 assay id Hs04260396_g1; CDKN1A assay id Hs00355782_m1; CDKN2A assay id Hs00923894_m1; bFGF (FGF2) assay id Hs00266645_m1; KGF (Keratinocyte Growth Factor, FGF7) assay id Hs00940253_m1; IGF-1 assay id Hs01547656_m1; and LEF1 assay id Hs01547250_m1.

TaqMan One-Step RT-PCR Master Mix Reagent (Thermo Fisher Scientific, Waltham, MA, USA) was used. The PCR reactions were performed on an ABI 7500 Real Time PCR system following the manufacturer’s instructions. The resulting data were analyzed with ABI software.

### 4.5. Western Blot Analysis

hDPCs (1 × 10^6^ cells/dish) were seeded in 100 mm dishes and cultured for 24 h. Adenosine was treated at concentrations of 0.75, 1.5, and 3 mM for an appropriate time. The cells were then lysed and total cellular proteins were prepared. The 50 μg protein samples were analyzed by western blotting with the corresponding antibodies; β-catenin (1:1000, Abcam, Cambridge, UK), GSK-3β (27C10) (1:1000, Cell Signaling Technology, Danvers, MA, USA), phospho-GSK-3β (Ser9) (1:1000, Cell Signaling Technology), an GAPDH (1:2000, Santa Cruz, Dallas, TX, USA). Western blot was analyzed by a chemiluminescence detector iBright FL1000 (Invitrogen, Waltham, MA, USA).

### 4.6. Protein Dot Blot Analysis for Human MAP Kinase Phosphorylation

Human MAP Kinase Phosphorylation Antibody Array Kit (Abcam, Cambridge, UK) was used to elucidate the changes in signal transduction pathways in DPCs. A total of 17 human proteins for phosphorylation were analyzed. Cells were treated with 1.5 and 3 mM of adenosine for an appropriate time and then collected for protein phosphorylation analysis. Cells treated with the vehicle medium were used as the non-treated control. The conventional immunoblot process was performed following the manufacturer’s protocol. The resulting blots were analyzed under the same condition using iBright FL1000 (Invitrogen, Waltham, MA, USA).

### 4.7. Intracellular cAMP Measurement

The intracellular cAMP level in cultured hDPCs was measured using the cAMP Assay Kit (Abcam, Cambridge, UK) following the manufacturer’s protocols. Cells were treated with adenosine and NECA for 2 min and harvested. Absorbance at 450 nm was measured using a microplate reader (BioTek, Winooski, VT, USA). Background wavelength correction was conducted at 540 nm.

### 4.8. Immunocytochemistry

hDPCs (8 × 10^4^ cells per well) were seeded in 24-well plates and cultured overnight. After washing with PBS, hDPCs were fixed with 4% paraformaldehyde at room temperature for 10 min. Cells were then permeabilized with PBS containing 0.1% Triton X-100 and blocked with PBS containing 5% FBS and 1% BSA. After consecutive incubation with primary antibodies (200:1 dilution, Abcam, Cambridge, UK) at 4 °C for 12 h and the Alexa 488 nm or Alexa 594 nm conjugated secondary antibodies (1000:1 dilution, ThermoFisher scientific, Waltham, MA, USA) at room temperature for 1 h, nuclei were stained with DAPI (2000:1 dilution, ThermoFisher scientific, Waltham, MA, USA) in the dark for 10 min. High resolution fluorescence images were taken using an EVOS^TM^ FL Auto2 Imaging System (ThermoFisher scientific, Waltham, MA, USA).

### 4.9. Statistical Analysis

All experimental data were presented as the mean ± standard deviation (S.D.) of at least three independent experiments. Experimental results were analyzed using SigmaPlot (Systat Software Inc., Chicago, IL, USA). The statistical significance of difference was determined by the Student’s *t*-test. The value of *p* < 0.05 was considered statistically significant.

## Figures and Tables

**Figure 1 molecules-27-02184-f001:**
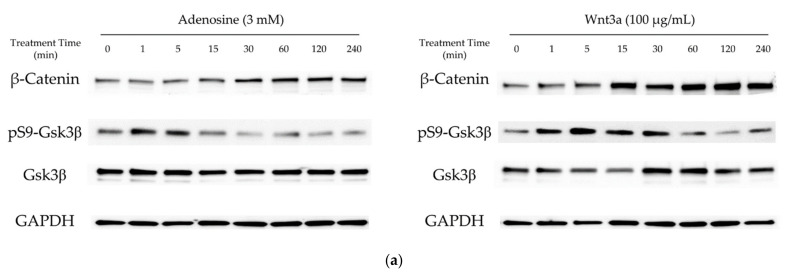
Adenosine activated Wnt/β-catenin signaling pathway in human dermal papilla cells. Cells were treated with adenosine for different times (0, 1, 5, 15, 30, 60, 120, and 240 min). (**a**) Whole cell lysates were analyzed by immunoblotting to determine the levels of β-catenin, pS9-Gsk3β, and Gsk3β. As an internal control, GAPDH was used. (**b**) The expression level of β-catenin, normalized with GAPDH, and the ratio of phosphorylated Gsk3β (Ser9) to total Gsk3β protein were evaluated. Wnt3a (100 ng/mL) was used as a positive control. (**c**) The mRNA expression profiles of ADORA1, ADORA2A, ADORA2B, and ADORA3 were examined in cultured human dermal papilla cells and HEK293A reporter cells. The data represent the means of six independent samples. Significantly different compared with 0 min treatment samples (β-Catenin * *p* < 0.05, ** *p* < 0.01, *** *p* < 0.001; ratio of pS9-Gsk3β/total Gsk3β (p-Gsk3β/t-Gsk3 β) # *p* < 0.05, ## *p* < 0.01, ### *p* < 0.001).

**Figure 2 molecules-27-02184-f002:**
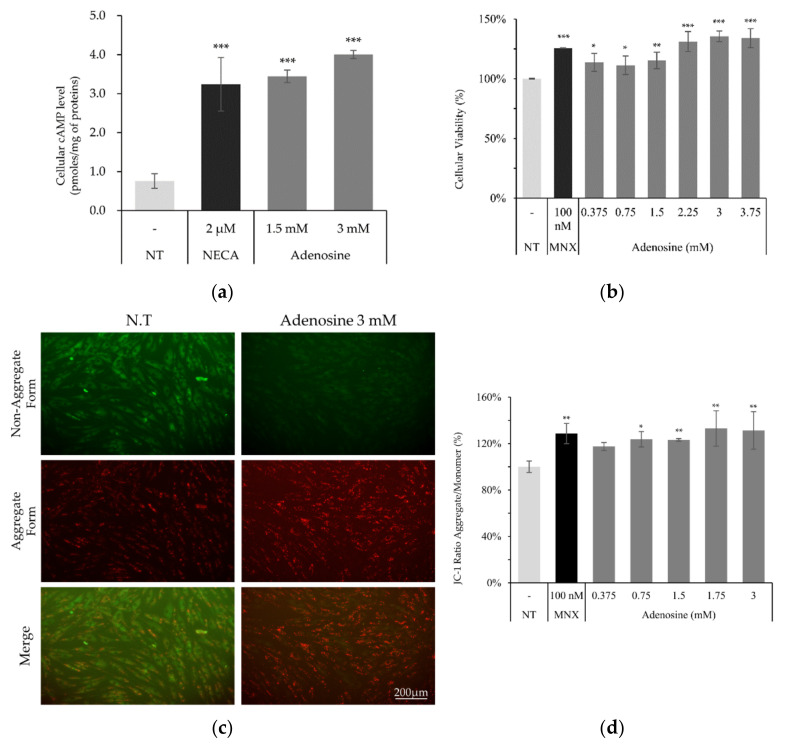
Adenosine enhanced cellular viability and mitochondrial energy metabolism in human dermal papilla cells. (**a**) Cellular cAMP level was increased by adenosine treatment for 2 min in hDPCs. (**b**) Cell viability measured by NADH/NADPH generation rates was enhanced by adenosine treatment for 4 h. (**c**) Mitochondrial membrane potential (△Ψ) was measured after adenosine treatment for 4 h. (**d**) JC-1 monomer form was seen as green and aggregate form as red by fluorescent microscopy. Minoxidil was used as a positive control. The data represent the means of six independent samples. MNX, minoxidil; N.T, non-treated control; Significantly different compared with N.T (* *p* < 0.05, ** *p* < 0.01, *** *p* < 0.001).

**Figure 3 molecules-27-02184-f003:**
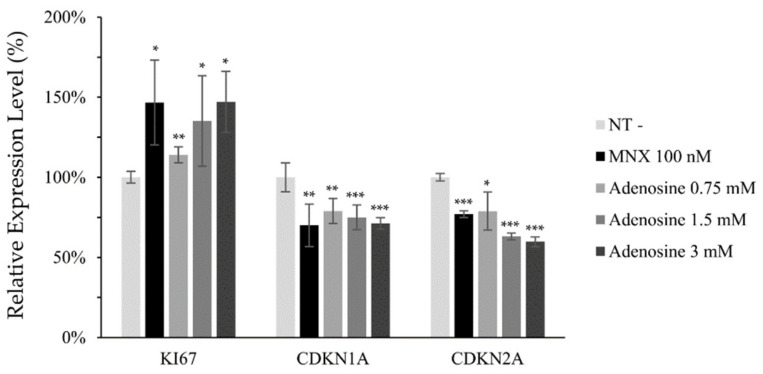
Adenosine modulated the mRNA expression of cell proliferation and cell cycle related genes in human dermal papilla cells. The mRNA expression level of cell proliferation marker (Ki67) and cell cycle arresters (CDKN1A and CDKN2A) was evaluated in hDPCs by RT-PCR after adenosine treatment for 24 h. Minoxidil was used as a positive control. The data represent the means of five independent samples. MNX, minoxidil; N.T, non-treated control; Significantly different compared with N.T (* *p* < 0.05, ** *p* < 0.01, *** *p* < 0.001).

**Figure 4 molecules-27-02184-f004:**
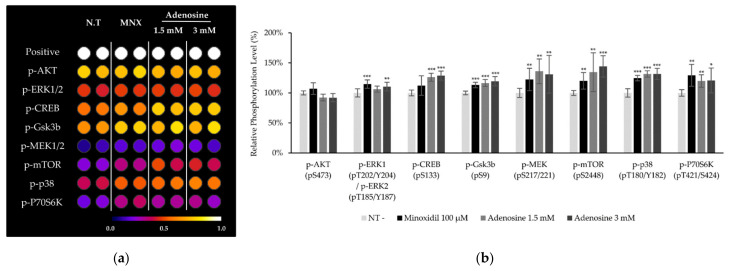
Adenosine enhanced protein phosphorylation and activated MAP kinases in human dermal papilla cells. (**a**) The phosphorylation of proteins including MAP kinases after 10 min treatment of adenosine was investigated. Seventeen proteins were analyzed and eight proteins are displayed. (**b**) Data were quantitated. Minoxidil was used as a positive control. The data represent the means of three independent samples. MNX, minoxidil; N.T, non-treated control; Positive, biotin-conjugated IgG; Significantly different compared with N.T (* *p* < 0.05, ** *p* < 0.01, *** *p* < 0.001).

**Figure 5 molecules-27-02184-f005:**
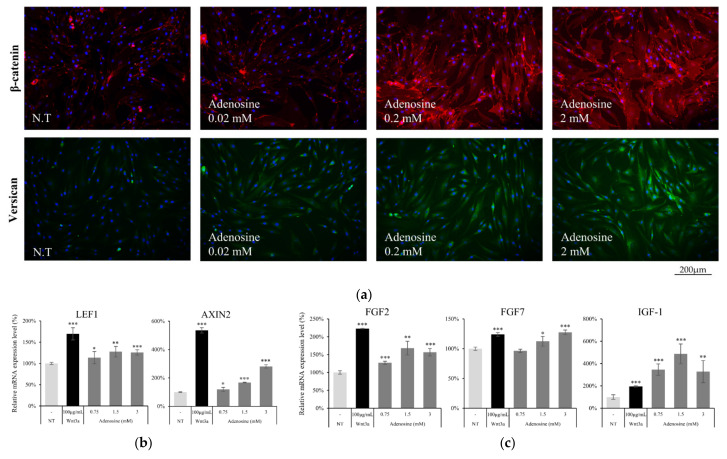
Adenosine activated Wnt target genes and growth factors in cultured human dermal papilla cells. (**a**) The induction of total β-catenin (**red**) and versican (**green**) expression level by 4 h treatment of adenosine was visualized by immunocytochemistry. Nuclei were stained by DAPI (**Blue**). The target genes of β-catenin were evaluated by RT-PCR after treatment with adenosine for 24 h. The mRNA expression level of (**b**) LEF1, AXIN2, and (**c**) growth factors (FGF2, FGF7, and IGF-1) and was increased by adenosine treatment. The data represent the means of five independent samples. NT, non-treated control; Significantly different compared with N.T (* *p* < 0.05, ** *p* < 0.01, *** *p* < 0.001).

**Figure 6 molecules-27-02184-f006:**
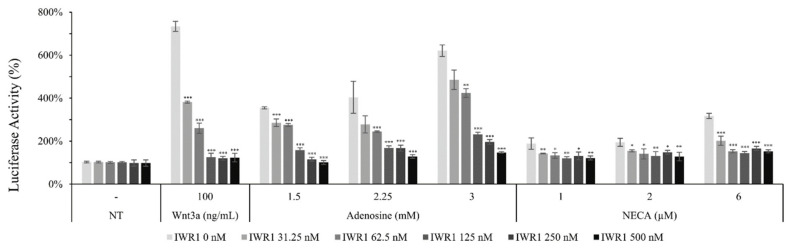
Adenosine stimulated canonical Wnt/β-catenin signaling pathway. The luciferase activity in WRHEK293A reporter cells was assessed after treatment with adenosine and NECA for 24 h in the presence/absence of IWR1, a Wnt inhibitor. Wnt3a was used as a positive control. The data represent the means of six independent samples. N.T, non-treated control; Significantly different compared with N.T (* *p* < 0.05, ** *p* < 0.01, *** *p* < 0.001).

**Figure 7 molecules-27-02184-f007:**
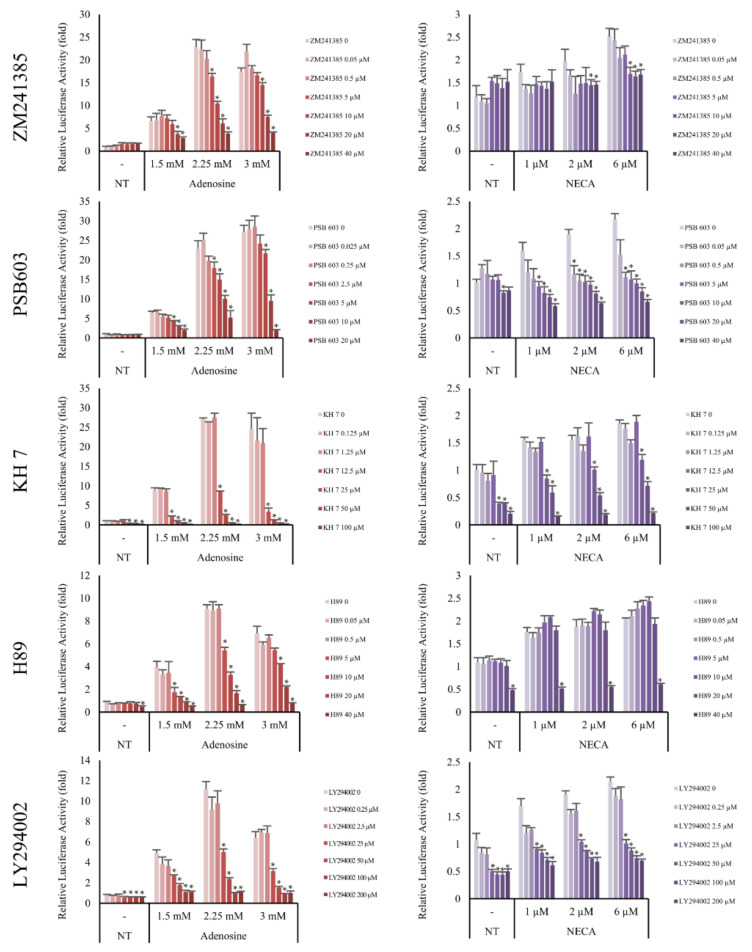
Adenosine A_2A_ and A_2B_ receptor antagonists inhibited adenosine mediated-Wnt/β-catenin signaling. The luciferase activity in WRHEK293A reporter cells was evaluated after treatment with adenosine and NECA for 24 h in the presence/absence of adenosine A_2A_ and A_2B_ receptor antagonists, ZM241385 and PSB603, respectively. Both inhibitors blocked Wnt/β-catenin reporter activity. The effects of the adenylyl cyclase inhibitor (KH7), PKA inhibitor (H89), and mTOR inhibitor (LY294002) were also examined. Adenosine-mediated Wnt activation was inhibited by KH7, H89, and LY294002. NECA was used as a positive control for adenosine A_2A_ and A_2B_ receptor activation. The data represent the means of six independent samples. N.T, non-treated control; Significantly different compared with N.T (* *p* < 0.001).

**Figure 8 molecules-27-02184-f008:**
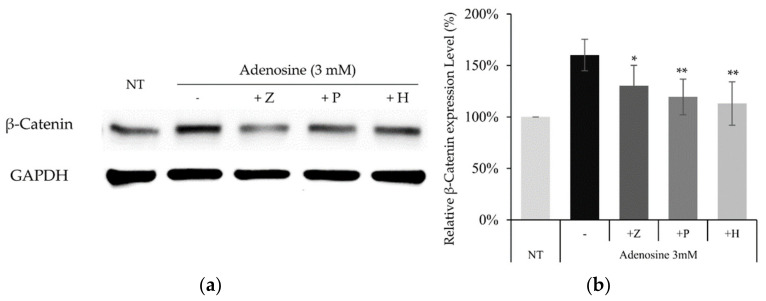
Activation of the Wnt/β-catenin pathway by adenosine was abolished in the presence of adenosine receptor antagonists and PKA inhibitor. The cells were treated with adenosine in the presence of various inhibitors (ZM241385, PSB603, and H89) for 4 h and harvested. (**a**) Whole cell lysates were analyzed by immunoblotting to determine the levels of β-catenin. GAPDH was used as an internal control. (**b**) The expression level of β-catenin was evaluated. The data represent the means of five independent samples. Significantly different compared with the adenosine treated control. (* *p* < 0.05, ** *p* < 0.01); Adenosine (3 mM); Z: ZM241385 (20 µM); P: PSB603 (10 µM); H: H89 (10 µM).

**Figure 9 molecules-27-02184-f009:**
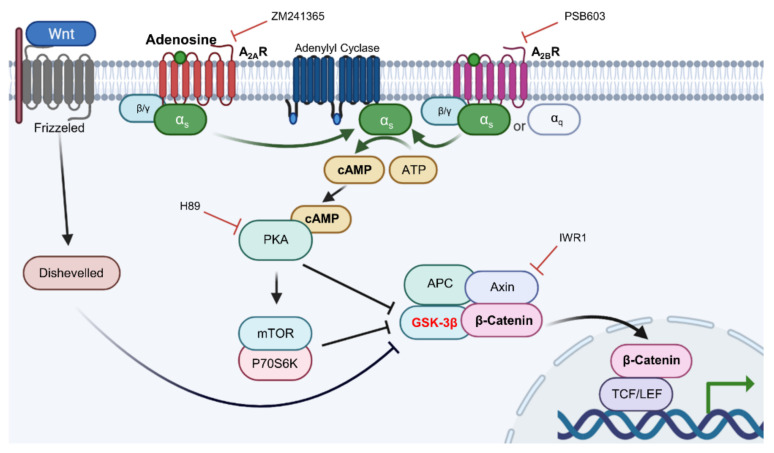
The proposed signal transduction pathways for adenosine-induced Wnt/β-catenin signal activation.

## Data Availability

The data presented in this study are available on request from the corresponding authors.

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
