# Peer review of "Anti-Hair Loss Effect of Adenosine Is Exerted by cAMP Mediated Wnt/β-Catenin Pathway Stimulation via Modulation of Gsk3β Activity in Cultured Human Dermal Papilla Cells"

_molecules, 2022, doi:10.3390/molecules27072184_

Round 1

Reviewer 1 Report

This study title as “Anti-Hair Loss Effect of Adenosine is Exerted by Camp 2 Mediated Wnt/β-catenin Pathway Stimulation via Gsk3β 3 Activity Modulation in Cultured Human Dermal Papilla Cells” aimed to investigate that adenosine stimulate Wnt/β-catenin signaling through the activation of adenosine receptor and Gsk3β plays critical role in transmitting the signals from adenosine receptor to β-catenin possibly via Gαs/cAMP/PKA/mTOR signaling cascade. This is an interesting and well-designed study to evaluate the functional pathway of adenosine. I have one minor concern about this study. Because the half-time of adenosine is very short, around 0.6 to 10 seconds. In this study, the authors evaluate the effects of adenosine for activating Wnt/β-catenin signaling pathway in human dermal papilla cells for different times (0, 1, 5, 15, 30, 60, 120 and 240 min). Different methodology for taking adenosine out from the sealing drug bottles might influence the finial results. That means if the technicians take too much time to take adenosine out from the drug bottles, the adenosine might not be effective and this study might not be repeated at other laboratories. The authors might address more details about their techniques for taking adenosine out from the drug bottles in the paragraph of methodology.   

Reviewer 2 Report

molecules-1630413

Anti-Hair Loss Effect of Adenosine is Exerted by cAMP Mediated Wnt/β-catenin Pathway Stimulation via Gsk3β Activity Modulation in Cultured Human Dermal Papilla Cells

This article focuses on the molecular mechanisms mediating hair growth promoting effects of adenosine in human dermal papilla cells (hDPCs). The authors report that cultured hDPCs predominantly expressed adenosine receptors A2B and A2A and that adenosine stimulation activated the Wnt/β-catenin signaling pathway. Moreover, adenosine enhanced the cell viability and the mitochondrial energy metabolism (as reflected by increased cAMP and NAD(P)H levels as well as elevated expression of markers of cell proliferation and mitochondrial activation). Adenosine also activated GSK3, p70S6K, mTOR, a variety of MAP kinases, transcription factor CREB, and various Wnt target genes. Further experiments indicated an involvement of adenosine receptors A2B and A2A as well as PKA in these effects. With regard to the mechanism of its hair growth promoting effect, the authors conclude that adenosine stimulates Wnt/β-catenin signaling via the receptors A2B/A2A and the subsequent inhibition of Gsk3β, a mechanism also including the activation of cAMP/PKA/mTOR-dependent pathways.

The study is well designed, carried out properly, and technically sound. In most parts, the presentation of the data is straightforward and clear. The results are predominantly convincing and conclusive. However, there are some critical points/questions requiring the authors’ consideration.

  1. Figure 1: Why is there a continuing (in part even increasing) stabilization of β-catenin at later time points (30-240 min) though the inhibitory GSK3β Ser9 phosphorylation is considerably reduced again?
  2. Please provide the number of independent experiments/replicates for all figures/panels.
  3. In the Results, please refer to the respective figures/panels showing the described data.
  4. Section 2.3: the molecules analyzed in this section should be described more precisely, e.g., GSK3, mTOR, and p70S6K are not MAPKs. Please also clarify which other MAPKs have been analyzed and whether they showed significant results.
  5. Figure 4: why are only some of the 17 MAP kinases analyzed shown?
  6. Figure 5b/c: to me, most of the adenosine-induced effects shown are rather small, i.e., less than 100% increase for LEF1, FGF2, and FGF7 (at all concentrations) as well as Axin2 mRNA (at 0.75 and 1 mM). To support the biological relevance of these results, significantly increased expression levels also have to be demonstrated on the protein level.
  7. In section 2.5 and the legend of Figure 7, the effect of Wnt3a as a positive control is described. In the figure, however, the use of Wnt3a is not indicated and I cannot see the respective data.
  8. Figure 7: the y axis has to be labelled. Please also discuss why H89 has such a small effect (i.e., only at the highest concentration) on luciferase activity in NECA-stimulated WRHEK293A reporter cells.
  9. In line 205, the authors refer to Figure 9 though Figure 8 is meant. Please correct.
  10. Discussion: a summarizing schematic figure should be included.
  11. The use of gene symbols has to be harmonized (e.g., FGF-7 vs. KGF).

Reviewer 3 Report

The paper is overall well written and the results are well planned and described.

There are some issues that deserve further attention and they are as follows:

  1. In Fig. 2a, the results relating to cAMP levels are reported only as a percentage of increase stimulated by various cell treatments. In my opinion, the values measured as pmoles/mg of protein or number of cells in untreated cells should be reported, in the text, at least, or in the legend.
  2. The Authors argue that the mechanisms of action of adenosine to induce its effects on DPCs are due to both cAMP and Wnt pathways. Since te relationship between these two paths is not so obvious, they should better substantiate the data supporting this possibility, citing relevant articles. Moreover, a scheme/figure could help the reader understand the interconnections between these two pathways. 
  3. A high dose of adenosine has been used in the experiments , as also highlighted in the Discussion, page 10, line 313. Could this dose, If administered in vivo, provoke side-effects? If yes, this fact should be mentioned , at least, and measures to limit this could be hypothesized.
  4. Minor points: the explanation for the acronym KFG is missing; on page 10, line 300, the word "inhibitors" is not entirely correct as between brackets two receptor antagonists and a PKA antagonist are reported and this should be appropriately indicated.

Round 2

Reviewer 2 Report

molecules-1630413

Anti-Hair Loss Effect of Adenosine is Exerted by cAMP Mediated Wnt/β-catenin Pathway Stimulation via Modulation of Gsk3β Activity in Cultured Human Dermal Papilla Cells

The manuscript provides a revised version of the manuscript “Anti-Hair Loss Effect of Adenosine is Exerted by Camp Mediated Wnt/β-catenin Pathway Stimulation via Gsk3β Activity Modulation in Cultured Human Dermal Papilla Cells”. The manuscript has been improved and my comments have been sufficiently addressed.

Reviewer 3 Report

The Authors have substantially satisfied the Reviewers' requests.

I noticed the following aspects that should be corrected without further revision:

  1. I requested to include baseline cAMP values, which were then increased by adenosine or NECA stimulation. The Authors  included two graphs, one with cAMP values expressed as pmoles/mg protein and the second with the percentage increase. I believe that the second is sufficient and the basal cAMP value in pmoles/mg of protein can be indicated in the legend or in the text of the Results. Otherwise, we would have two graphs with similar results.
  2. The English should be revised since there are still errors (i. e.  in the legend of Fig. 4, "17 proteins were analyzed" is confusing and should be "17 protein samples were analyzed"; at page 12, line 289, "and the inhibitory phosphorylation of ... were suggested" should be "and the inhibitory phosphorylation of ... was suggested" and so on) but I wish that the editorial staff can help the Authors in correcting all text.
  3. As for the sentence about the poor side effects resulting from adenosine administration in the scalp, I would conclude that " Based on these considerations, the topically applied adenosine should not provoke significant systemic adverse effects but this aspect deserves further investigation".